# Human Breast Milk miRNAs: Their Diversity and Potential for Preventive Strategies in Nutritional Therapy

**DOI:** 10.3390/ijms242216106

**Published:** 2023-11-09

**Authors:** Bertrand Kaeffer

**Affiliations:** Nantes Université, INRAE, UMR 1280, PhAN, F-44000 Nantes, France; bertrand.kaeffer@univ-nantes.fr or bertrand.kaeffer@inrae.fr

**Keywords:** nutritional programming, neonate, miR-26, miR-320

## Abstract

The endogenous miRNAs of breast milk are the products of more than 1000 nonprotein-coding genes, giving rise to mature small regulatory molecules of 19–25 nucleotides. They are incorporated in macromolecular complexes, loaded on Argonaute proteins, sequestrated in exosomes and lipid complexes, or present in exfoliated cells of epithelial, endothelial, or immune origins. Their expression is dependent on the stage of lactation; however, their detection depends on progress in RNA sequencing and the reappraisal of the definition of small RNAs. Some miRNAs from plants are detected in breast milk, opening the possibility of the stimulation of immune cells from the allergy repertoire. Each miRNA harbors a seeding sequence, which targets mRNAs, gene promoters, or long noncoding RNAs. Their activities depend on their bioavailability. Efficient doses of miRNAs are estimated to be roughly 100 molecules in the cytoplasm of target cells from in vitro and in vivo experiments. Each miRNA is included in networks of stimulation/inhibition/sequestration, driving the expression of cellular phenotypes. Three types of stress applied during lactation to manipulate miRNA supply were explored using rodent offspring: a foster mother, a cafeteria diet, and early weaning. This review presents the main mature miRNAs described from current mothers’ cohorts and their bioavailability in experimental models as well as studies assessing the potential of miR-26 or miR-320 miRNA families to alter offspring phenotypes.

## 1. Introduction

Milk is a complex food produced by the mammary gland, providing a baby with essential nutrients for growth along with an immunity kit for short-term survival and long-term epigenetic information influencing the health of the future adult. During lactation, the composition of milk is tuned to the baby’s needs from colostrum at birth up to the end of lactation at weaning [1]. Milk from different mammals can be used to feed a human baby, suggesting that molecules like miRNAs—known to retain biological activity from flies to mammals—can easily be tested on a relevant animal model for potential preventive use in the nutritional therapy of infants. Many ribo- and desoxyribonucleic acids of complex food are recycled by the digestive system. However, some escape digestion and are used in cell-to-cell signaling or are provided by immune or exfoliated epithelial cells directly to the offspring. A mature miRNA is a single-stranded ribonucleic acid of 19–25 nucleotides in length, which is generated by the RNase-III-type enzyme Dicer from an endogenous transcript that contains a local hairpin structure [2,3]. The miRNAs are encoded either within the introns of protein-coding genes or transcribed under the control of their own promoters. After transcription as long primary transcripts, they are trimmed into hairpin intermediates (pre-miRNAs) in the nucleus, subsequently exported to the cytoplasm, then cleaved into mature miRNAs. The hairpins usually code for 3p and 5p mature miRNAs with differential tissular expressions [4]. The miRNAs function as post-transcriptional repressors of their target genes when bound to specific sites in the 3′ untranslated region (UTR) of the target mRNA. The binding of the miRNAs relies on ‘seed pairing’, i.e., the perfect or near-perfect complementary match of nucleotides 2–8 of the mature miRNA product [5]. In silico database exploration has predicted binding sites on the promoter 5′-UTR coding domain of mRNA as well as on lnc-RNA. The equivalence of let-7a binding sites in 5′-UTR or 3′-UTR for repression has been demonstrated by transfecting HeLa cells [6]. miRNAs can be detected in the cytoplasm, nucleus, nucleolus [7], and mitochondria [8]. Noncoding ribonucleic acids (small or long ncRNA) are involved in epigenetic regulation. These molecules directly silence or activate chromatin at specific loci or through their integral role in the machinery that drives DNA methylation. Breast milk contains miRNAs in high amounts [9]; these have been proposed to act as epigenetic regulators [10,11,12,13].

The review presents the main mature miRNAs described across current mothers’ cohorts and the conditions of their bioavailability in experimental models as well as studies assessing the potential of miR-26 or miR-320 miRNA families to alter offspring phenotypes.

## 2. RNA Content and Mature miRNA Diversity

The miRNA composition of breast milk has been explored by RNA sequencing performed with kits designed with an RNA extraction step [14] or without [15], as in a recent breast milk cohort [16]. q-PCR is used for the specific exploration of known miRNAs and confirmation studies, with normalization provided by reference endogenous genes like miRNAs or by spiking samples with xenogenous miRNA like cel-lin4-5p.

In clinical practice, breast milk is classified as a noninvasive sample available in maternity wards or collected at home on a worldwide scale. The main parameters to ponder when designing a clinical plan are the mother’s delivery (term or preterm), the capacity of mammary glands to deliver the food (difference between breasts; the compositional variation during suckling between fore, middle, and hind milk; and the circadian rhythm of production) (Figure 1A). To control for the differences between breasts, mothers are sometimes asked to use the same breast for sampling at each time-point. Minimal differences in miRNA content have been found between fore and hind milk [17]. It has been advised that pre-feed samples should be utilized in order to minimize confounding [18]. Minimal impacts of freeze–thaw cycles have been found on milk miRNAs [19,20], which is not surprising as the milk of domestic animals retains cryoprotective properties. All these parameters have to be taken into account when designing a sampling procedure [21].

In clinical trials, available volumes varying between 100 µL and 100 mL have been reported (Figure 1B). Although q-PCR is a sensitive method, the design of a breast milk analysis using 100 µL (down to 50 µL [20]) is challenging, but it is a prerequisite when exploring milk samples from the same mother during a nychthemeron or when sharing samples between multiple analyses (transcriptomic, lipidomic, and metabolomic [22]). We have to consider that the sampling of a single drop of breast milk clearly has less informative value than a volume of over 5 mL. Multiplexed techniques to allow bulk extractions from low volumes of crude milk are needed in the field.

The RNA content of whole breast milk after classical phenol/chloroform extraction is wide; for instance, 90 to 1000 ng/µL from 8 parturients [23]. Breast milk contains a high amount of small or very small noncoding RNAs [24] along with transfer RNA, messenger RNA, and long noncoding RNA [25,26].

**Figure 1 ijms-24-16106-f001:**
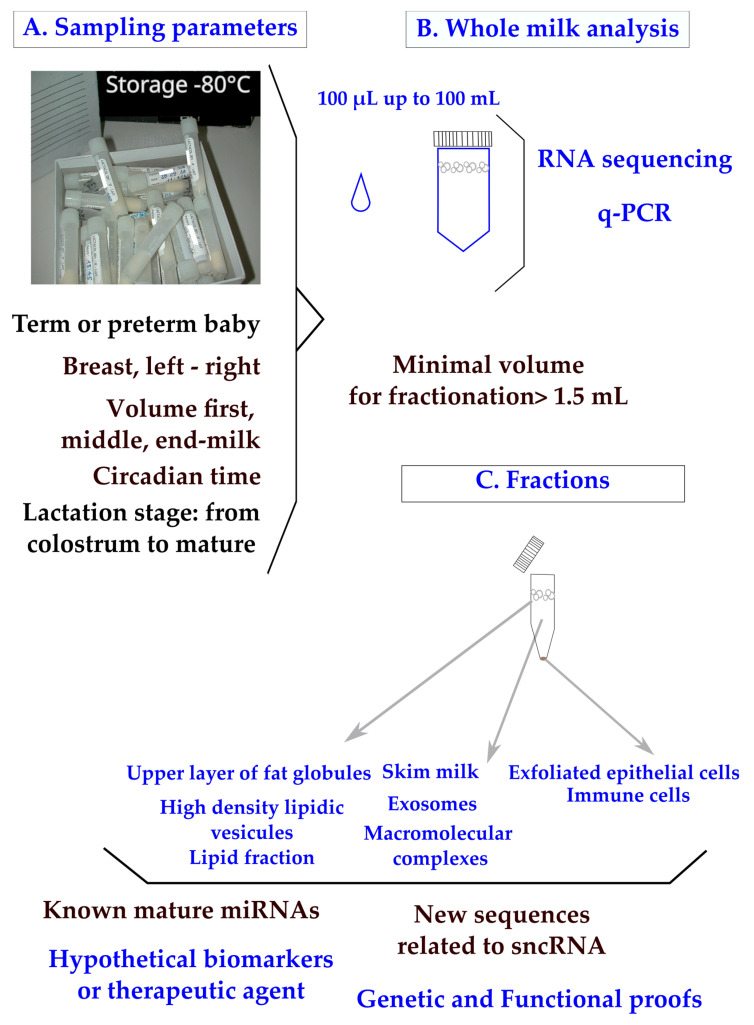
Human breast milk composition: sampling parameters, analyses of whole milk, and fractions. Sampling parameters of whole milk (**A**) were explored using RNA sequencing or q-PCR. (**B**) Clinical samples pose a major problem of legal pressure to reduce the sample volume. Milk fractions have been explored down to 1 mL [27] and 1.5 mL of whole milk [19]. (**C**) Recommendations have been made to centrifuge breast milk immediately after sampling to remove exfoliated cells, then to store them at −80 °C or, preferably, under liquid nitrogen. A step at −20 °C is frequently applied when mothers are sampled at home.

In the lipid fraction of breast milk (Figure 1C), the variations in known and newly described miRNAs have been explored using mothers with normal diets [18,28,29] or comparatively with mothers consuming high-fat diets [30]. Many short RNAs are undescribed and, beyond as an interest as biomarkers, new regulatory processes may be discovered when exploring the total content of whole milk. The classification of miRNAs is designed on biogenesis [2,31] with four criteria: (1) the confirmation of expression by hybridization to a size-fractionated RNA sample; (2) the small RNA sequence has to be present in one arm of the hairpin precursor, which lacks large internal loops or bulges; (3) the small RNA sequence has to be phylogenetically conserved and the sequence conservation has to be seen in the precursor hairpin, but to a lesser extent than in the mature miRNA segment; and (4) the evidence is stronger if the precursor accumulates in the presence of a reduced Dicer function. More recently, the definition has been reassessed to distinguish the function from the biogenesis of miRNAs from other classes of RNA by dissociating the three notions of an miRNA gene (Mir), an miRNA precursor (pre-MiRNA), and a mature miRNA product (MiRNA). Under that assumption, miRNA genes produce mature miRNAs, but some miRNAs may originate from genes transcribed into other types of noncoding RNAs [32]. These definitions are of importance when performing RNA sequencing with the aim of discovering new miRNAs.

The recovery of exosomes is influenced by the procedure of sampling (Figure 1C) and a systematic centrifugation step of the milk before storage has been proposed in order to remove cells and debris [33]. A minimal volume of 1.5 mL whole milk has been used to purify exosomes [19], which is a technical lock, up to higher volumes like 20 mL [34].

The diversity of miRNAs has been related to the stage of lactation. A strikingly different composition of miRNAs between colostrum and mid-lactation from 18 parturients has been found (seven miRNAs were confirmed by q-PCR: miR-511-3p, 429, 29c-3p, 885-5p, 30b-5p, 183-5p, and 623 [35]). Previously, a different composition of miRNAs comprising one colostrum against five breast milk has been reported. miR-518c-3p miRNAs is uniquely detected in breast milk [9] and is also expressed in the placenta with a putative use as a preeclampsia biomarker [36]. Seven miRNAs (miR-148a-3p, 22-3p, 26a-5p, 21-5p, 7b-5p, 7g-5p, and 24-3p) have been found to be common to nipple aspirate, serum, plasma, breast tissue, and breast milk [37]. Surprisingly, the lists have no common miRNAs; this is notable in the let-7 family, which is highly expressed in tissues [37] and used as reference gene [20]. let-7g miRNAs are considered to be a reliable molecular method for body-fluid identification markers at crime scenes [38]. It should be underlined that the levels of microRNAs of the let-7 family in exosomes from mothers suffering from type 1 diabetes are widely divergent from normal levels [39].

Many studies on whole milk have attempted to derive a specific set of miRNAs common to all mothers at a specific stage of lactation or related to the mother’s pathological status. A list of the most highly expressed breast milk families of miRNAs has been proposed from the meticulous analysis of 19 studies as follows: the let-7-5p family, miR-30-5p family, and miR-146b-5p, used as reference genes in q-PCR; the miR-148a-3p, miR-200a-3p, miR-141-3p, miR-22-3p, miR-181-5p families; miR-378a-3p; the miR-29-3p family; and, lastly, miR-200b/c-3p [40].

The content of known miRNAs can be related to immune function, epigenetics, insulin regulation, and growth. According to the analysis of current milk formulations, miR-148a-3p and miR-125b-5p are highly expressed [41] and are considered to target the gastrointestinal tract [10].

Studies agree on the high expression of miR-146a-5p, which has been linked to food allergies. The human milk oligosaccharide 2′-fucosyllactose is known to moderate β-lactoglobulin-induced food allergies through the miR-146a-mediated toll-like receptor 4/nuclear factor-κB signaling pathway [42].

An analysis of whole milk revealed that the expression of miR-146b-5p, which is not abundant in plasma, was abundantly expressed in breast milk [29,40,43]. miR-146b-5p has been found to be crucial in the homeostasis of alveolar mammary glands [44].

An exploration of the epigenetic properties of breast milk miRNAs was conducted on miR-148a-3p and on the families of miR-26 and miR-320, which are frequently reported in analyses but are in lower amounts. In cell monolayers incubated with human milk-derived miRNA from the skim and fat layers, the expression of DNA methyltransferase 1 (dntm1), a target gene of miR-148a, was downregulated with an upregulation of miR-148a, a result predicted by comparative functional genomics [11]. The importance of these microRNAs for lactation is frequently reported, particularly for miR-148a-3p, which has been consistently described as the most abundant microRNA in different milk fractions (fat, whey, and extracellular vesicles) [45]. However, in milk of mothers suffering from gestational diabetes mellitus, the expression of miR-148a-3p is lower than in normal milk [46]. The miR-148a-3p expressions in the fat and skim layers of human milk are equivalent, just like for bovine and goat fractions. The expression of miR-148a-3p is significantly lower in infant formula than in human milk [47].

The maternal diet modulates the levels of specific miRNAs in human milk lipid fractions [30], most notably for miR-148a-5p and miR-146b-5p as well as maternal weight [48,49]. miR-26a-5p has also been described in another cohort as a lipid milk fraction [29]. However, the impact of the maternal environment on specific changes to milk miRNA content and the potential involvement in the metabolic programming of the offspring are not fully understood. The human breast milk supply of specific miRNAs is affected by maternal overweight/obesity and may influence the infant body mass index [49].

hsa-miR-26a-5p MIMAT0000082, UUCAAGUAAUCCAGGAUAGGCU, is highly fsimilar to 26b-5p, just like hsa-miR-26a-1-3p MIMAT0004499, 5′-CCUAUUCUUGGUUACUUGCACG-3, is highly similar to 26a-2-3p and 26b-3p. The family is widely expressed in mammals [50].

Among the newly detected miRNAs in whole milk, the isoform e of the miR-320 family [16] is strongly in favor of a wide expression of the miR-320 family (a to e). The miR-320 family contains five members in humans (a, b, c, d, and e), all from 3p. hsa-miR-320-3p is identical in rats (MIMAT0000903) and mice (MIMAT0000666). In humans, it is mainly known as 320a (or 320-3p) (5-AAAAGCUGGGUUGAGAGGGCGA-3; MIMAT0000510). The 18 nucleosides in the five members are identical to all families and constitute an e form; the 3 end of b is GCAA, c is GU, and d is A. This miRNA does not harbor a hexanucleotide element that putatively facilitating an address to the nucleus. However, on exosomes purified from the breast milk of a population from the Faroe Islands, miR-320e-3p was described in one of the four clusters [51], suggesting that more studies are needed to delineate the importance of ethnicity. On milk samples collected from 38 healthy mothers (either with preterm (n = 15) or full-term infants (n = 23)), miRNA-320-3p was more highly expressed in the colostrum of full-term infants than in the milk of mothers with preterm infants [52]. The expression of miRNA-148 was higher in preterm mother’s milk than full-term colostrum. The expression of miRNA-320-3p and miRNA-148a were found to be upregulated in cells incubated with milk exosomes [52], which led to a decrease in the target genes fatty acid synthase 1 and DNA methyltransferase 1, respectively. hsa-miR-148a-5p (MIMAT0004549; 5′-AAAGUUCUGAGACACUCCGACU-3′) does not share any sequence similarity with the miR-26 or miR-320 families.

Up to now, the number of studies on the composition of noncoding RNA is relatively modest according to the population of women of reproductive age, ethnic diversity, and the diets in use on a world scale. The consequence is that discrepancies between studies cannot properly be explained according to the huge diversity of environmental factors like food or mother’s ethnicity. Due to the high connectivity of miRNAs, new integrative strategies between omics and the clinical parameters of the mother have to be designed to evidence existing logical links. The studies on miR-26 and miR-320 families illustrate that more work under the redundancy hypothesis of action of near-identical molecules or paralogs are needed for proper therapeutic use. The next section presents the data available on the conditions of miRNA bioavailability in the digestive tract.

## 3. Bioavailability of Mature miRNAs in the Digestive System

The systemic RNA interference–deficient transporter (sidt1) is the main receptor of dietary and orally administered miRNAs in the digestive system [53], indicating that miRNAs present in breast milk can transfer to offspring. However, miR-375-3p cannot cross the digestive tract of mice [54]; even when using the breast milk of a mouse engineered to produce a high amount of miR-30b, a demonstration of plasma loading was impossible [55].

As a consequence, the bioavailability of miRNA depends on sheltering these molecules from the molecular environment. It determines the capacity of miRNA molecules to reach target cells with a concentration high enough to trigger a biological response, a crucial step in digestive fluids highly loaded with RNases. In vitro data on cell cultures have shown that a ratio of 100 miRNA molecules delivered to the target cell cytoplasm triggers a measurable physiological response [56]. In vivo, after a rough estimation of the total cells of a rat gastric mucosa to adapt to a concentration of miR-320-3p or miR-375-3p, we were able to confirm that this ratio could be used in an oral gavage [57,58].

Milk is rich in exosomes, which can serve as natural cargo for miRNAs [59]. However, these molecules are frequently widely present in all milk fractions. Thus, fluorescently labeled miRNAs like miR-375-3p must be transfected in exosomes in order to show accumulations in the liver, spleen, and brain after suckling or oral gavages. A demonstration was conducted using mice and by transfecting bovine exosomes with fluorescent miRNA administered to mice [60]. Another approach is to take advantage of the milk composition related to a certain pathology of the mother. Human milk exosomes from gestational diabetes mellitus (GDM) and healthy parturients showed distinct regulatory bioactivities in both HepG2 cell cultures and, in vivo, in the liver of Balb/c mice. The profile of GDM exosomes has been related to natural loading by miR-101-3p [34]. Previous studies confirmed that the target gene of miR-101-3p is mTOR. miR-101-3p suppresses mTOR by binding in the 3′-UTR regions of mRNAs [61].

Works on milk exosomes are a very active field of investigation because (1) a food source of a single miRNA species to supplement diets with a crude product does not exist; (2) exosomes can protect RNA from digestive enzymes; and (3) they can be tailored by the genetic engineering of cell lines to load miRNAs with molecular data, addressing relevant cells or tissues [62]. Some natural miRNAs can be addressed to exosomes [63] with a step of loading the miRNA onto Argonaute proteins before loading into exosomes [64].

However, miRNA obtained by chemical synthesis can also be loaded into artificial vectors like lipoaminoglycoside Dioleyl-Succinyl Paromomycin [65]. The loading of miR-375-3p has been measured using a transgenic rat model on the enteroendocrine cell lineage [58]. Taking this specific miRNA as an example, miR-375-3p has been proposed as a key regulator in malignant breast cancer [66]. In cohorts of breast-fed infants, the consumption of miR-375-3p was associated with protection from atopy [67], opening the consideration of the prevention of allergic diseases through breastfeeding. The levels of miR-375-3p were found to be upregulated in the breast milk of mothers treated with probiotics, although the study concluded that this miRNA and others could not be considered to convey protection against allergic diseases to the baby [68]. These divergent properties illustrate the problem caused by supplying miRNA in an animal model; most of the time, the site of delivery impacted different cellular phenotypes. Another level of complexity is added if one considers that the immune cells involved in allergy may be also stimulated by miRNAs from plants present in breast milk [23]. As the chemistry of plant miRNAs is different from eucaryotes, complicating the analysis of breast milk, the presence of these xenogenous molecules has to be taken into account in future allergy studies [69].

The oral administration of miRNAs is a current problem in targeted nutritional therapy and requires further studies, both to design new prokaryotic or eukaryotic vectors [70,71] and to test miRNA cocktails. A specific problem with models of rodent babies is to ensure that the stomach is empty by separating the pups from their mother in one hour before gavage [57,58]. The biological relevance of miRNAs in low amounts is counterintuitive as a high abundance frequently correlates with a high bioactivity [72]. But, the continued uptake of milk-derived exosomes that carry dnmt-targeting miRNAs may promote diabetes, allergies [69], neurodegenerative diseases, and cancer later in life [11].

A hypothesis driven by an in silico approach related miR-148a-5p to dnmt1 regulation, opening the possibility that the composition of breast milk could manipulate the homeostasis of tissues like the pancreas [73]. Targeted nutritional therapy requires not only screening miRNAs for potential epigenetic activity, but also designing natural or biomimetic vectors, addressing the load of molecules to a specific cell lineage. In the last section, studies addressing the long-term consequences of manipulating breast milk miRNAs are presented.

## 4. Potential of miRNAs for In Vivo Treatment of Offspring

Gestation and lactation constitute a unique window of opportunity to supplement babies to minimize the negative effects of early nutritional programming [74], among which is obesity [75]. During early postnatal life, stressful experiences can lead to long-term programming with deleterious consequences on adult health. The application of preventive and therapeutic approaches during specific windows of opportunity like hospitalization in intensive care units holds promise for the improvement of the future health of the next generations. For example, if the epigenetic patterns disrupted by the exposition to stress can be modified through specific epigenome-targeted therapeutic interventions, it would be possible to correct the impaired profile of gene expressions to prevent stress-induced chronic pathologies and improve human health and longevity. Three types of stress applied during lactation to manipulate miRNA supply have been explored using rodent offspring. The effect of a foster mother in a model using a/a and Avy/a mice with a C57Bl6J background examined the consequences on first and second generations and survival parameters as end-points as well as oocyte metabolism (Figure 2A [76]). The other studies discussed here were limited to the first generation. A cafeteria diet consumed by lactating mothers decreased the level of miR-26a-3p, with consequences on the F1 generation (Figure 2B [77]). Likewise, the effects of an oral supply of miR-320-3p given to early-weaned rat pups on the expression of polr3d in digestive epithelia or brain cells was explored over the long term (Figure 2C [58]).

Alternative breastfeeding practices allowing human milk sharing, such as cross-fostering or donor-milk banking, are used in hospitals [78] or for convenience. A mouse model allowed researchers to follow the expression patterns of miR-186-5p, miR-141-3p, miR-345-5p, and miR-34c-5p and their candidate target genes Mapk8, Gsk3b, and Ppargc1a in ovarian and liver tissues [76]. The main result related miR-186-5p to the glycogen synthase kinase 3 beta (gsk3b) gene in F2 ovaries, a negative regulator of glucose homeostasis involved in energy metabolism, inflammation, ER stress, mitochondrial dysfunction, and apoptotic pathways. Defects in this gene have been associated with Parkinson’s disease and Alzheimer’s disease. The miR-186-5p expression decreased in F2 milk siblings, whereas Gsk3b increased compared with the control counterparts. These data on breast milk were in line with those of a pilot study conducted to analyze the miRNA profiling of gastric contents from both breast-fed and formula-fed infants [79,80]. The gastric content of infants at autopsy were analyzed, demonstrating the presence of miRNAs in the gastric content. Differences in the miRNA content regarding the type of feeding (breast-fed or formula-fed) have been reported, leading to the proposal of miR-151a and miR-186-5p as potential breast milk biomarkers [79]. These two studies [76,79] pave the way for the exploration of the long-term effects of miR-186-5p supply in transgenerational rodent models.

miR-26a-5p, an abundant miRNA in breast milk, has been linked to adipogenesis in offspring. Thus, milk miR-26a may act as an epigenetic regulator influencing the early metabolic program of the progeny. This miRNA could be seen as a relevant component for correct development in optimized milk formulations. miR-26 suppresses murine adipocyte progenitor cell differentiation and functions within the adipocyte progenitor cell lineage to inhibit mobilization and subsequent adipocyte production by downregulating FBXL19, a novel driver of adipogenesis [81]. After weaning, descendants of cafeteria-fed dams breastfed with lower levels of miR-26a displayed greater expressions of Hmag1, Rb1, and Adam17 in retroperitoneal white adipose tissue in comparison with controls. Hence, alterations to the amount of miR-26a supplied through milk during lactation could alter the expression of target genes in murine descendants and may affect adipose tissue development [77].

**Figure 2 ijms-24-16106-f002:**
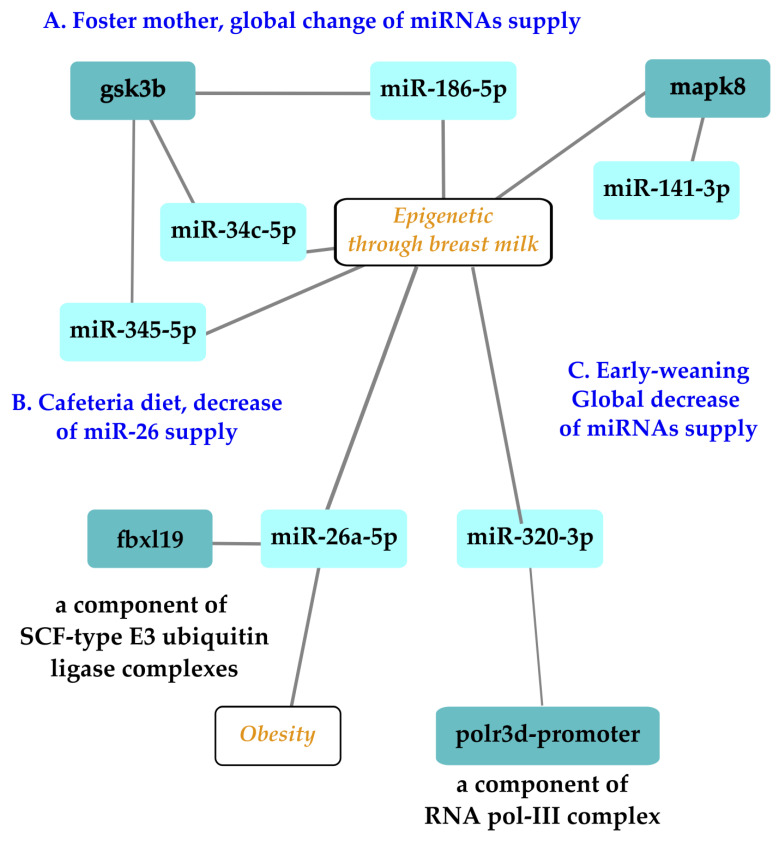
Exploration of three types of stress related to miRNAs with epigenetic properties delivered during lactation on the first or second offspring generations. Experiments on a murine model of foster mothers showed that miRNAs like miR-186-5p could be impacted at the oocyte level in offspring, opening the possibility of an impact on first and second offspring generations (**A**). Independent experiments showed that a cafeteria diet altered the miR-26 family, with consequences on the adipogenesis of the first generation (**B**). An oral supplementation of pups with miR-320-3p during lactation had an immediate effect on digestive cells and a long-term impact on the brain compartments (**C**). Note that miR-26 and -320 families have several isoforms, which can all be redundant molecules with a similar bioactivity. Nomenclature: glycogen synthase kinase 3 beta, gsk3b, polr3d promoter, RNA polymerase III subunit 3d promoter, F-box, and leucine-rich repeat protein 19 (fbxl19). Illustration drawn with Cytoscape [82].

In breast milk, miR-320-3p is less expressed in milk from mothers with a preterm infant than a full-term delivery [52], suggesting that a supplementation would be beneficial to the health of the preterm baby. miR-320, an endogenous shRNA, is highly expressed in mammals [83]. The in vivo delivery of miR-320-3p targeted binding sites located both on the promoter of RNA polymerase III and on the 3′-UTR of the transcript [57,58,84]. Polr3d is subunit 17 of RNA polymerase III, which is involved in human tumorigenesis. RNA polymerase III is considered to be linked to aging and longevity through TORC and insulin genes as well as through genes related to telomerase activity. Additional work is needed to explore the putative long-term effects on the polr3d complex, which includes 17 subunits, as well as any effect on telomerase activity [85,86]. Up to now, the oral supplementation of rat pups by miRNA-320-3p or miR-375-3p during lactation has revealed long-term miRNA-specific consequences on the endogenous levels of corresponding miRNAs with a strong tissue-dependent memory. Combining an miR-320-3p treatment with an early weaning schedule showed a strong interaction between miRNA treatments and early weaning in all tissue extracts, except the brain stem. In the hippocampus, miR-504 was downregulated in both sexes, but in the brain stem, it was upregulated only in females, along with the miR-320-3p and miR-16-5p levels. In the hypothalamus, Clock was upregulated in both sexes.

Future explorations need to relate the behavior of rodents submitted to miRNA treatment during lactation in order to widen the long-term perspectives of such preventive strategies [58]. The aminoglycoside vector is efficient when delivering miRNAs to the cytoplasm of all digestive cells without loading the gastric exosomes [57,58]. Such vectors allow for the administration of a cocktail of several RNA molecules, which constitutes a promising field of future exploration.

Clearly, for therapeutic applications, an effect on offspring gonads [76] is not desired. Moreover, the influence of a single miRNA on multiple tissues [58] is also a difficult problem to address in studies in the prevention of metabolic diseases.

## 5. Conclusions and Perspectives

The concept of Developmental Origins of Health and Disease (DOHaD) highlights a critical period of development during the first 1000 days of life, including the period from conception to the end of the second year of life. That period is probably the most active for epigenetic regulation, especially for DNA imprinting [86]. Much evidence supports the theory that early nutrition affects epigenetic programming, which triggers adult disease over the long term [87].

RNAs are able to interact with DNAs, RNAs, and proteins, building up biochemical networks within a single cell or connecting multiple cell types. Extensive validation studies are needed to evaluate the side-effects of RNA-based drugs [88] as well as the reappraisal of selective pressure on genome regulation machinery [89] as detailed descriptions of the competing endogenous RNA network (ce-RNA) underpinning immune cell regulation. However, the ce-RNA hypothesis has been challenged by a work on miR-122 [90], demonstrating that solutions to eradicate severe pathologies under this hypothesis are unlikely. Nevertheless, there is an exciting possibility of preventing ophthalmic diseases in the next generation by reprogramming heritability in mice; thus, hypoxic stress of the retina could prepare offspring for a beneficial response to such treatments [91].

Designing a new milk formula taking into account the epigenetics of the child [92] allows the unification of the molecular and global points of view of epigenetics [93]. Small transcoded RNAs (microRNA, pi-RNA, etc.) play important roles in embryo development by influencing various mRNA expressions [94] as well as in transgenerational epigenetic inheritance. As both maternal and paternal lineages influence epigenetic alterations in offspring [95], the design of an optimized milk formulation could prove to be challenging for future research.

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
