# Peer review of "Human Breast Milk miRNAs: Their Diversity and Potential for Preventive Strategies in Nutritional Therapy"

_ijms, 2023, doi:10.3390/ijms242216106_

Round 1

Reviewer 1 Report

Comments and Suggestions for Authors

The authors presented a review summarizing mature miRNAs found in current maternal cohorts, explored their bioavailability in experimental models, and examined studies investigating the potential effects of miR-26 and miR-320 miRNA families on altering offspring phenotype.

While the topic is intriguing, I believe that the current manuscript requires some minor improvements before it is ready for publication.

 1.     The section (lines 45-49) discussing miRNA binding to target genes requires extension. I recommend providing more experimentally confirmed examples of miRNA binding within the 5’UTR and CDS regions of mRNA. The sentence “The miRNAs biding relies mainly on “’seed pairing” should be revised.

2.     Lines 160-163 should include a specific explanation of the similarities among the four mentioned miRNAs. Additionally, it would be valuable to provide the sequences for all four of these miRNAs.

3.     Figure 1,2 should be improved in terms of quality and resolution. Adjust the font size and color to enhance readability, and aim to make the Figure 1 more visually appealing and comprehensible to readers. It currently appears more like typed text rather than a clear figure.

Author Response

Reviewer-1

The authors presented a review summarizing mature miRNAs found in current maternal cohorts, explored their bioavailability in experimental models, and examined studies investigating the potential effects of miR-26 and miR-320 miRNA families on altering offspring phenotype.

While the topic is intriguing, I believe that the current manuscript requires some minor improvements before it is ready for publication.

I thank the Reviewer-1 for the positive appreciation, and I am offering corrections with justification below.

 1.     The section (lines 45-49) discussing miRNA binding to target genes requires extension. I recommend providing more experimentally confirmed examples of miRNA binding within the 5’UTR and CDS regions of mRNA. The sentence “The miRNAs biding relies mainly on “’seed pairing” should be revised.

Answer-1 : The section has been modified to provide examples of experimentally confirmed miRNA binding in 5’-UTR on experiments made by Lytle et al., 2007.

The sentence about miRNAs binding is now «The miRNAs biding relies on ‘seed pairing’.

The miRNAs binding relies on ‘seed pairing’ (the perfect or near-perfect complementary match of nucleotides 2–8 of the mature miRNA product [5]. In silico database exploration predict binding sites on promoter, 5’UTR, coding domain of mRNA, as well as on lnc-RNA. The equivalence of let-7a binding sites in 5’-UTR or 3’-UTR for repression has been reported by transfecting HeLa cells [6]. These molecules can be detected in cytoplasm, nucleus, nucleolus [7], and mitochondria [8].

2.     Lines 160-163 should include a specific explanation of the similarities among the four mentioned miRNAs. Additionally, it would be valuable to provide the sequences for all four of these miRNAs.

Answer-2 : The sequences of the miRNAs are now quoted in the text.

The hsa-miR-26a-5p MIMAT0000082, « UUCAAGUAAUCCAGGAUAGGCU » is highly similar to 26b-5p ; just like the hsa-miR-26a-1-3p MIMAT0004499, “5’-CCUAUUCUUGGUUACUUGCACG-3” is highly similar to 26a-2-3p and 26b-3p. The family is widely expressed in mammals [51].

Among the newly detected miRNA in whole milk, the isoform e of the miR-320 family [16], strongly in favor of a wide expression of miR-320 family (a to e). The miR-320 family contains 5 members in human (a, b, c, d, e) all from 3p. The hsa-miR-320-3p is used here because of sequence identity in rat (MIMAT0000903) and mouse (MIMAT0000666). In human it is mainly known as 320a (or 320-3p) (5 -AAAAGCUGGGUUGAGAGGGCGA-3 ; MIMAT0000510). The 18 nucleosides in 5 are identical to all family and constitute e form; the 3 end of b is “GCAA,” of c “GU” and of d “A.” This miRNA does not harbor hexanucleotide element putatively facilitating addressing to nucleus. However, on exosome purified from breast milk of a population of the Faroe island, the miR-320e-3p is described in one of the 4 Clusters [52], suggesting that more studies are needed to delineate the importance of ethnicity. On milk samples collected from 38 healthy mothers from preterm (n = 15) and full-term infants (n-23), the MiRNA-320-3p was more highly expressed in the colostrum of fullterm than in the milk of mothers with preterm infants [53]. The expression of MiRNA-148 was higher in preterm mother's milk than of full-term colostrum. MiRNA-320-3p and MiRNA-148a expression were up-regulated in cells incubated with milk exosomes [53], which lead to a decrease in their target genes Fatty acid synthase 1 and DNA methyltransferase 1, respectively. The hsa-miR-148a-5p, (MIMAT0004549 , 5’-AAAGUUCUGAGACACUCCGACU-3’) is not sharing any sequence similarity with the miR-26 or miR-320 families.

3.     Figure 1,2 should be improved in terms of quality and resolution. Adjust the font size and color to enhance readability, and aim to make the Figure 1 more visually appealing and comprehensible to readers. It currently appears more like typed text rather than a clear figure.

Answer-3 : The figures are submitted with a higher resolution. I have drastically improved the Figure-1.

Reviewer 2 Report

Comments and Suggestions for Authors

The author provides overview of miRNA expression patterns in breast milk and their potential modulation during lactation, with specific emphasis on their role in offspring programming. The miRNA families identified in this review may represent new targets for interventions aimed at promoting healthy outcomes for breastfed infants. There some comments to the author.

Comments:

1. How plants miRNAs are transmitted into breast milk should be more evidence.

2. I suggest that the author include the problems and their respective solutions regarding the oral administration of miRNAs.

3. Each miRNA has multiple target genes, and identifying breast milk miRNAs for nutritional therapy is very challenging. It would be more convincing if more examples could be provided in this regard.

4. The font size is inconsistent and needs to be revised.

5. The subtitle 3 on page 7 needs revision as the words are concatenated together.

Comments on the Quality of English Language

 Minor editing of English language required

Author Response

Reviewer-2

The author provides overview of miRNA expression patterns in breast milk and their potential modulation during lactation, with specific emphasis on their role in offspring programming. The miRNA families identified in this review may represent new targets for interventions aimed at promoting healthy outcomes for breastfed infants. There some comments to the author.

I thank the Reviewer-2 for positive comments made for improving the manuscript. I am proposing corrections with justification below.

Comments:

1. How plants miRNAs are transmitted into breast milk should be more evidence.

Answer-1: Data are available on plant miRNAs bioavailability and consequences on the infant immunology. I have added the review of “Acevedo N, Alashkar Alhamwe B, Caraballo L, Ding M, Ferrante A, Garn H, Garssen J, Hii CS, Irvine J, Llinás-Caballero K, López JF, Miethe S, Perveen K, Pogge von Strandmann E, Sokolowska M, Potaczek DP, van Esch BCAM. Perinatal and Early-Life Nutrition, Epigenetics, and Allergy. Nutrients. 2021 Feb 25;13(3):724. doi: 10.3390/nu13030724. PMID: 33668787; PMCID: PMC7996340”.

These divergent properties illustrate the problem paused by supplying miRNA in animal model, most of the time, the site of delivery will impact differently cellular phenotypes. Another level of complexicity is added if one considers that immune cells involved in allergy may be stimulated also by miRNAs from plant present in breast milk [23]. As the chemistry of plant miRNAs are different from eucaryotes, complicating the situation of breast milk analysis, the presence of these xenogenous molecules have to be taken more into account in futures allergy studies [69].

2. I suggest that the author include the problems and their respective solutions regarding the oral administration of miRNAs.

Answer-2: I have included details about oral administration of miRNAs directing toward our own protocols.

Oral administration of miRNAs is a current problem in targeted nutritional therapy and requires further studies both to design new prokaryotic or eukaryotic vectors [67, 68] and to test miRNA cocktails. A specific problem with rat model is to ensure that the stomach is empty by separating the pups from their mother during one hour, before gavage [55, 56].

3. Each miRNA has multiple target genes, and identifying breast milk miRNAs for nutritional therapy is very challenging. It would be more convincing if more examples could be provided in this regard.

Answer-3. The field is new and only three studies have explored the consequences of manipulating miRNAs content on the next generation.

4. The font size is inconsistent and needs to be revised.

Answer-4: Font size has been checked.

5. The subtitle 3 on page 7 needs revision as the words are concatenated together.

Answer-5: Subtitle now stands as: ‘3. Potentials of miRNAs for in vivo treatment of offspring’

Comments on the Quality of English Language

 Minor editing of English language required.

Answer to ‘Quality of English Language’. The manuscript has been proofread to remove minor editing problems.

Round 2

Reviewer 2 Report

Comments and Suggestions for Authors

The authors have been replied the comments point to point. I have no more comment.

Author Response

I am grateful for your comments, clearly improving my manuscript.

I am answering below all points.

There are several instances where the text moves from human to animal models and back without a clear delineation.  Some areas in the text where this is evident are
i). paragraph 211-218  This is a rat model, but no one would know it.

Indeed, the text is modified:

In vivo, after a rough estimation of the total cells of the rat gastric mucosa to adapt the concentration of miR-320-3p or miR-375-3p, we confirm that this ratio can be used in oral gavage [57,58]”.

ii). paragraph 219-222 (Ref. 60)- bovine, porcine and murine models, but again not clear it isn't human.

I have improved the text to avoid ambiguity.

However, these molecules are frequently widely present in all milk fractions. So fluorescently labeled miRNAs like miR-375-3p must be transfected in exosomes in order to show accumulation in liver, spleen, and brain after suckling or oral gavage. The demonstration has been done in mice and by transfecting bovine exosomes with this fluorescent miRNA administered to mice [60].”

iii). paragraph 302-312 (Ref. 82) is mouse studies. Ref. 78- studies of rodents and the 3T3 cell line.

I introduced the information below.

The miR-26 suppresses murine adipocyte progenitor cell differentiation and functions within the adipocyte progenitor cell lineage to inhibit mobilization and subsequent adipocyte production by down-regulating FBXL19, a novel driver of adipogenesis [81]. After weaning, descendants of cafeteria-fed dams breastfed with lower levels of miR-26a displayed greater expression of Hmag1, Rb1, and Adam17 in retroperitoneal white adipose tissue in comparison with controls. Hence, alterations in the amount of miR-26a supplied through milk during lactation is able to alter the expression of target genes in the murine descendants and may affect adipose tissue development [77]”.

iv). lines 322-329- not sure if this is human or rodent.

I have introduced the species, human for tumorigenesis, but rat pups for the demonstration of Tavares et al., 2023.

Polr3d is the subunit-17 of RNA polymerase-III involved in human tumorigenesis. RNA polymerase III is considered to be linked to aging and longevity through TORC and insulin genes, as well as through genes related to telomerase activity. Additional work is needed to explore the putative long-term effects on the polr3d complex, which includes 17 subunits, as well as any effect on telomerase activity [85, 86]. Up to now, oral supplementation of rat pups by miRNA-320-3p or miR-375-3p during lactation has long-term miRNA-specific consequences on the endogenous levels of corresponding miRNAs with a strong tissue-dependent memory.”